# Mechanical Behaviors of Polymer-Based Composite Reinforcements within High-Field Pulsed Magnets

**DOI:** 10.3390/polym16050722

**Published:** 2024-03-06

**Authors:** Siyuan Chen, Tao Peng, Xiaotao Han, Quanliang Cao, Houxiu Xiao, Liang Li

**Affiliations:** 1Wuhan National High Magnetic Field Center, Huazhong University of Science and Technology, Wuhan 430074, China; chen_siyuan@hust.edu.cn (S.C.); pengtao@mail.hust.edu.cn (T.P.); xthan@hust.edu.cn (X.H.); quanliangcao@hust.edu.cn (Q.C.); xiaohouxiu@mail.hust.edu.cn (H.X.); 2State Key Laboratory of Advanced Electromagnetic Engineering and Technology, Huazhong University of Science and Technology, Wuhan 430074, China

**Keywords:** pulsed magnet, polymer-based composites, structural performances, damage and failure mechanism, Zylon fiber

## Abstract

The development of pulsed magnets capable of generating magnetic fields exceeding 100 Tesla has been recognized as a crucial pursuit for advancing the scientific research on high magnetic fields. However, the operation of magnets at ultra-high magnetic fields often leads to accidental failures at their ends, necessitating a comprehensive exploration of the underlying mechanisms. To this end, this study investigates, for the first time, the mechanical behaviors of Zylon fiber-reinforced polymers (ZFRPs) within pulsed magnets from a composite perspective. The study begins with mechanical testing of ZFRPs, followed by the development of its constitutive model, which incorporates the plasticity and progressive damage. Subsequently, in-depth analyses are performed on a 95-T double-coil prototype that experienced a failure. The outcomes reveal a notable reduction of approximately 45% in both the radial and axial stiffness of ZFRPs, and the primary reason for the failure is traced to the damage incurred by the end ZFRPs of the inner magnet. The projected failure field closely aligns with the experiment. Additionally, two other magnet systems, achieving 90.6 T and 94.88 T, are analyzed. Finally, the discussion delves into the impact of transverse mechanical strength of the reinforcement and axial Lorentz forces on the structural performance of magnets.

## 1. Introduction

Pulsed magnets are the exclusive devices capable of generating repeated magnetic field intensity exceeding 50 T within milliseconds. They play an essential role as fundamental scientific research tools in the fields of condensed matter physics and magnetics [1,2,3]. The ability of magnets to manipulate the electronic states of matter is significantly enhanced with increasing magnetic field intensity. Consequently, research laboratories, such as the National High Magnetic Field Laboratory (NHMFL), Dresden High Magnetic Field Laboratory, and Wuhan National High Magnetic Field Center (WHMFC), have dedicated the past four decades to achieving magnetic field intensities of 100 T or higher [4,5,6]. The current world record stands at 100.75 T, achieved by NHMFL [7], while WHMFC attained 94.88 T in 2021 [6]. Both of them have set their sights on reaching 110 T in the future.

Nevertheless, the upgrading of pulsed magnets poses considerable challenges due to the substantial Lorentz forces involved. Nowadays, pulsed magnets employ an inner reinforcing structure, where each conductor layer is reinforced with fiber composites, as depicted in Figure 1a,b. Since the radial component predominantly contributes to the Lorentz forces (especially on the mid-plane of magnets), composites are designed with a winding angle of 90° and manufactured by customized winding machines (Figure 1c). Although this approach results in magnets with low axial stiffness and strength, it remains substantial for low-field operations. Nonetheless, as the field intensity increases, the axial forces induced by the radial stray field become significant (especially on the magnet ends), frequently leading to inadvertent damage. The composites at the magnet ends are found to undergo complete destruction, accompanied by dislocation of the wires [3,8,9,10]. The underlying failure mechanism remains unclear, and effective countermeasures have yet to be identified.

The unclear failure mechanism primarily arises from the inadequate analyses of composites within pulsed magnets. Traditionally, the composite reinforcements are considered as ideal anisotropic elastic materials, employing the von Mises failure criterion, and a design is deemed reasonable if the von Mises stress of the reinforcement remains below 80% of the fiber-direction ultimate tensile strength (UTS) [11,12,13]. However, this treatment overlooks the transverse and shear failure modes, as well as the nonlinear behaviors exhibited by composite materials. It should be noted that, during high-field operations, the reinforcement encounters radial and axial compression stresses exceeding 200 MPa, thereby increasing the likelihood of transverse damage [14,15,16]; however, the von Mises criteria would obscure this underlying damage. Additionally, laboratories currently adopt Zylon fibers as reinforcement fibers due to their exceptional strength and excellent electrical insulation. Nevertheless, Zylon fibers display poor impregnation with epoxy, resulting in a notable plastic behavior [17,18,19]. Hence, it is insufficient to solely consider the ideal anisotropic elasticity.

Moreover, the lack of systematical testing on Zylon fiber-reinforced polymers (ZFRPs) poses another problem. Significant research efforts have been dedicated to investigation of the longitudinal ultimate tensile strength of ZFRPs [20,21,22,23]. Only Huang et al. conducted measurements regarding the transverse compression strength of ZFRPs with cylindrical and bar-shaped specimens, but these results exhibited notable dispersion due to stress concentration effects [23]. As a result, the transverse and in-plane shear properties of ZFRPs remain insufficiently understood, impeding efforts to elucidate the underlying failure mechanism.

Taking all these factors into account, this paper presents, for the first time, a comprehensive analysis on the mechanical behavior of ZFRP reinforcements from a composite perspective. The study begins with comprehensive mechanical testing of ZFRPs, followed by the development of a plasticity-damage model. Subsequently, the simulation technique for pulsed magnets is proposed, and the mechanical behaviors of ZFRP reinforcements inside three double-coil magnet systems are thoroughly analyzed. These systems include a 95-T prototype that experienced failure, as well as two magnets achieving magnetic fields of 94.88 T and 90.6 T, respectively. Finally, the influence of the transverse mechanical properties of reinforcement and axial Lorentz forces on the structural failure are discussed.

## 2. Materials and Methods

In general, the ZFRP reinforcements within pulsed magnets are wound under high pre-stress, achieving a high fiber filling factor (*V_f_*) of 0.8. However, due to the challenges encountered in manufacturing a standard specimen with such a high *V_f_*, specimens with a moderate *V_f_* of 0.53 were initially tested. Subsequently, the mechanical properties of ZFRPs with *V_f_* of 0.8 were deduced based on a bridging model. The mechanical tests have been conducted in our recent work [24] but will still be outlined in this section for clarity.

### 2.1. Specimen Preparation

To begin with, a Zylon prepreg was manufactured using a one-step hot melt method. A thermoset epoxy with a moderate characteristic curing temperature was selected as the resin matrix. The epoxy exhibited a tensile elastic modulus of 3.2 GPa and UTS of 70 MPa, and its flexural strength and modulus were measured as 130 MPa and 1.07 GPa, respectively.

Subsequently, ZFRP laminates were produced using the autoclave-forming method. The temperature was gradually raised to 350 K at a rate of 2 K/min and held for 30 min. Next, the temperature was further increased to 400 K and maintained for 90 min, followed by a natural cooling process to room temperature. The entire process was conducted under a pressure of 0.8 MPa. Eight layers of prepreg were employed, resulting in a laminate with a thickness of 2 mm, while the *V_f_* was determined to be 0.53.

Furthermore, an E-glass end tab was attached to the laminate using a low-temperature structural adhesive. The adhesion areas were polished and cleaned with acetone. To mitigate the stress concentration during transverse tension testing, wedge-shaped tabs were incorporated (according to ASTM D3039 [25]). Finally, laminates were machined into the desired shape using a diamond band saw system (EXKAT300CL). To mitigate the fluffing of Zylon fibers, the ZFRP laminate was sandwiched between two carbon/epoxy laminates during the cutting process.

### 2.2. Mechanical Tests

The tensile tests of ZFRPs were conducted under ASTM D3039, while the compression and in-plane shear tests were based on ASTM D6641 [26] and ASTM D3518 [27], respectively. The testing results are presented in Table 1. For clarity, the notation used is as follows: ‘1’ denotes the fiber direction; ‘2’ signifies the transverse direction; ‘*c*’ denotes compression properties while ‘*t*’ signifies tension properties; ‘*E*’ denotes elastic modulus (‘*G*’ denotes the shear modulus in special); ‘*X*’ signifies strength; and ‘0′ denotes the elastic limit strength. It should be noted that, due to the limitations of the environmental chamber, the results of 77 K were obtained through linear fitting or extrapolation.

The stress value $X_{22}^{\;t,0}$ in Table 1 was defined as the stress at which the deviation from the linear zone reached 0.5 MPa, while the stress value $X_{12}^{\;t,0}$ was defined as the stress at which the attenuation of the secant in-plane shear modulus reached 7%. In addition, tension tests on symmetric laminates ${\left(\pm 55{}^{\circ}\right)}_{2s}$ and ${\left(\pm 67.5{}^{\circ}\right)}_{2s}$ were performed. Their tension strengths at 293 K were determined to be 32.6 MPa and 22.4 MPa, respectively. The tested stress–strain curves will be employed to verify the validity of material model in Section 3.3.

### 2.3. Properties with High V_f_

Based on the testing results, the properties of Zylon fiber at 77 K were deduced as $E_{11}^{f}$ = 327 GPa, $E_{22}^{f}$ = 2.7 GPa, $G_{12}^{f}$ = 1.9 GPa, and $G_{23}^{f}$ = 1.3 GPa, according to the bridging model [26]. The bridging model utilizes the Mori–Tanaka approach to establish a relationship between the stresses in the fiber and the matrix, enabling the calculation of the strength and modulus of composites with a high degree of accuracy [28]. Subsequently, the bridging model was employed to predict the mechanical properties of ZFRPs with a *V_f_* of 0.8. The predicted results are listed in Table 2 and will be used in the subsequent sections.

## 3. Constitutive Modeling of ZFRPs

To characterize the nonlinearity of ZFRPs caused by the internal damages and inherent plasticity, an anisotropic model of FRPs constructed in our previous work was employed [29]. The effectiveness of the model was demonstrated through biaxial-tension and open-hole-tension tests of glass and carbon FRP laminates. This section will outline the constructed model and verify its applicability to ZFRPs.

### 3.1. Plasiticity Evolution

An anisotropic elastoplastic model was built to capture the plasticity evolution. The effective engineering stress and elastic strain are represented in vector form as follows:(1)$$\tilde{\sigma }={\left[{\tilde{\sigma }}_{11},{\tilde{\sigma }}_{22},{\tilde{\sigma }}_{33},{\tilde{\sigma }}_{12},{\tilde{\sigma }}_{23},{\tilde{\sigma }}_{13}\right]}^{T}$$
(2)$$\varepsilon^{e}={\left[\varepsilon_{11}^{e},\varepsilon_{22}^{e\;},\varepsilon_{33}^{e\;},\varepsilon_{12}^{e\;},\varepsilon_{23}^{e\;},\varepsilon_{13}^{e\;}\right]}^{T}$$

Based on the assumption of strain equivalence, these two quantities follow the relationship of
(3)$$\dot{\tilde{\sigma }}=Μ\cdot \dot{\sigma }=diag\left(\frac{1}{1-d_{1}},\frac{1}{\eta_{E}\left(1-d_{2}\right)},\frac{1}{\eta_{E}\left(1-d_{2}\right)},\frac{1}{\eta_{G}\left(1-d_{6}\right)},\frac{1}{\eta_{G}\left(1-d_{6}\right)},\frac{1}{\eta_{G}\left(1-d_{6}\right)}\right)\cdot \dot{\sigma }=Μ\cdot D^{e}\cdot {\dot{\varepsilon }}^{e}$$
where $D^{e}$ denotes the Voigt form of the four-order stiffness tensor; **M** signifies the damage matrix; $d_{1}$, $d_{2}$, and $d_{6}$ correspond to the coefficients reflecting the fiber-direction, transverse, and shear micro-damages, respectively; and $\eta_{E}$ and $\eta_{G}$ correspond to the coefficients reflecting the transverse and shear macro-damages, respectively. The evolution of *d* and *η* will be introduced in Section 3.2.

The equivalent stress is formulated as follows [30]:(4)$${\bar{\tilde{\sigma }}}_{y}=\sqrt{\frac{3}{2}\left[{\left({\tilde{\sigma }}_{22}\right)}^{2}+{\left({\tilde{\sigma }}_{33}\right)}^{2}+2a{\left({\tilde{\sigma }}_{12}\right)}^{2}+2a{\left({\tilde{\sigma }}_{13}\right)}^{2}+2a{\left({\tilde{\sigma }}_{23}\right)}^{2}\right]}$$
where *a* represents the coupling between transverse plasticity and in-plane shear plasticity. The associated yield function and equivalent plastic strain are given by:(5)$$\Phi =\frac{1}{2}{\tilde{\sigma }}^{T}P\tilde{\sigma }-{\bar{\tilde{\sigma }}}_{y}^{2}({\bar{\varepsilon }}^{p})$$
and
(6)$${\bar{\tilde{\sigma }}}_{y}({\bar{\varepsilon }}^{p})=\beta {\left({\bar{\varepsilon }}^{p}\right)}^{\alpha }$$
where **P** is defined as diag(0,3,3,6*a*,6*a*,6*a*). The material-related coefficients *a*, *α*, and *β* are determined as 2.0, 0.3, and 900 for ZFRPs through in-plane shear tests. The initial yield stress is set as zero considering that ‘*α*’ is tiny. Details of the determination process of these coefficients can be found in [29].

The flowing vector **N** of the equivalent strain is
(7)$$N=\frac{\partial \Phi }{\partial \tilde{\sigma }}=P\cdot \tilde{\sigma }$$

Thus far, the equivalent plastic strain can be expressed as
(8)$${\dot{\bar{\varepsilon }}}^{p}=2\dot{\gamma }{\bar{\tilde{\sigma }}}_{y}=\sqrt{\frac{2}{3}{\left({\dot{\varepsilon }}_{22}^{p}\right)}^{2}+\frac{2}{3}{\left({\dot{\varepsilon }}_{22}^{p}\right)}^{2}+\frac{1}{3a}{\left({\dot{\varepsilon }}_{12}^{p}\right)}^{2}+\frac{1}{3a}{\left({\dot{\varepsilon }}_{23}^{p}\right)}^{2}+\frac{1}{3a}{\left({\dot{\varepsilon }}_{13}^{p}\right)}^{2}}$$
where *γ* is the Lagrange’s plastic multiplier. Accordingly, the hardening law is
(9)$${\dot{\bar{\varepsilon }}}^{p}=\dot{\gamma }H=\dot{\gamma }\sqrt{N^{T}\cdot Z\cdot N}$$
where *H* is the hardening modulus and **Z** is defined as diag(0,2/3,2/3,1/3*a*,1/3*a*,1/3*a*).

### 3.2. Damage Evolution

Puck’s criteria were employed to determine the material damage. Considering that the ZFRP reinforcements of magnets are subjected to radial, axial, and hoop stresses, the three-dimensional (3D) form of Puck’s criteria were utilized. Details of Puck’s criteria can be found in [31]. Necessary coefficients of Puck’s criteria were listed in Table 2.

The Cachan continuum damage model was employed to describe the evolution of micro-damages [32,33,34]. The damage strain energy density *W_D_* is redefined in terms of the action-plane stresses as
(10)$$W_{D}=\frac{1}{2}\left[\frac{\sigma_{11}^{2}}{E_{11}}-\frac{2\upsilon_{11}}{E_{11}}\sigma_{11}\sigma_{n}+\frac{{\langle \sigma_{n}\rangle }_{+}^{2}}{E_{22}\left(1-d_{2}\right)}+\frac{{\langle \sigma_{n}\rangle }_{-}^{2}}{E_{22}}+\frac{\tau_{nt}^{2}+\tau_{n1}^{2}}{2G_{12}\left(1-d_{6}\right)}\right]$$

The corresponding damage-development equations are [30,31]:(11)$$\left\{ \begin{array}{l} d_{2}=\frac{{\langle \sup\left(\sqrt{Y_{d^{\prime }}(t)+bY_{d}(t)}\right)-Y_{0}\rangle }_{+}}{Y_{c}}\\ d_{6}=\frac{{\langle \sup\left(\sqrt{Y_{d^{\prime }}(t)+bY_{d}(t)}\right)-Y_{0}^{\prime }\rangle }_{+}}{Y_{c}^{\prime }} \end{array}\right.$$
with
(12)$$\left\{ \begin{array}{l} Y_{d}=\frac{\partial W_{D}}{\partial d_{2}}=\frac{1}{2}\frac{{\langle \sigma_{n}\rangle }_{+}^{2}}{E_{11}{\left(1-d_{2}\right)}^{2}}\\ Y_{d^{\prime }}=\frac{\partial W_{D}}{\partial d_{6}}=\frac{1}{2}\frac{\tau_{nt}^{2}+\tau_{n1}^{2}}{G_{12}{\left(1-d_{6}\right)}^{2}} \end{array}\right.$$

The values of *b*, $Y_{0}$, $Y_{c}$, $Y_{0}^{\prime }$, and $Y_{c}^{\prime }$ were determined as 12.7, 0.21, 6.16, 0.3, and 4.16 for ZFRPs, through cyclic tensile tests of ${\left(\pm 45{}^{\circ}\right)}_{2s}$ and ${\left(\pm 67.5{}^{\circ}\right)}_{2s}$ laminates. The determination process of these coefficients can be found in [30].

Furthermore, the macro-damages were characterized using the macro-phenomenon method [35], which is given as Equation (13). In this equation, ‘0′ signifies the tangent modulus at the onset of macro-damages and ‘*f*_E,IFF_’ indicates Puck’s stress exposure for inter-fiber fracture. Since the stiffness of ZFRPs degrades rapidly after the emergence of macro-damages, the parameters *c*, *η*, and *ξ* were set as 100, 0.01, and 0.8, respectively.
(13)$$\left[\begin{array}{l} E_{22}\\ G_{12} \end{array}\right]=\left[\begin{array}{l} E_{33}\\ G_{23} \end{array}\right]=\left[\begin{array}{l} \eta_{E}\cdot E_{22}^{0}\\ \eta_{G}\cdot G_{12}^{0} \end{array}\right]=\left[\begin{array}{l} \left(\frac{1-\eta_{rE}}{1+c_{E}{\left(f_{E,IFF}-1\right)}^{\xi_{E}}}+\eta_{rE}\right)\cdot E_{22}^{0}\\ \left(\frac{1-\eta_{rG}}{1+c_{G}{\left(f_{E,IFF}-1\right)}^{\xi_{G}}}+\eta_{rG}\right)\cdot G_{12}^{0} \end{array}\right]$$

### 3.3. Numerical Implementation

The constitutive model was implemented in Fortran and integrated into the ANSYS 19.0 software by the UserMat feature. The role of UserMat is to update the Cauchy stress and consistent tangent stiffness by utilizing the received stress and strain at every material integration point during the solution phase.

For the Cauchy stress, the return mapping algorithm was employed. In a plastic loading condition, the elastoplastic Equations (1)–(9) are simplified to
(14)$$\Phi \left(\Delta \gamma \right)=\frac{1}{2}{\left[{\left(I+\Delta \gamma D^{e}P\right)}^{-1}\cdot {\tilde{\sigma }}_{n+1}^{trial}\right]}^{T}\cdot P\cdot \left[{\left(I+\Delta \gamma D^{e}P\right)}^{-1}\cdot {\tilde{\sigma }}_{n+1}^{trial}\right]-{\bar{\tilde{\sigma }}}_{y}^{2}\left({\bar{\varepsilon }}_{n+1}^{p,trial}+\Delta \gamma H\left({\tilde{\sigma }}_{n+1}\right)\right)$$

The only unknown variable Δγ can be calculated by Newton–Raphson iteration and the stress and damage state can be evaluated. For the consistent tangent stiffness $D^{ep}$, a chain derivation was employed and the stiffness can be deduced according to Equation (15).
(15)$$D^{ep}=M^{-1}\cdot {\left[{\left(D^{e}\right)}^{-1}+\Delta \gamma P-\frac{\Delta \gamma \cdot \left(NN^{T}\right)\cdot \left(ZP\right)}{H^{2}}+\frac{NN^{T}}{2H{\bar{\tilde{\sigma }}}_{y}\alpha \beta {\left({\bar{\varepsilon }}^{p}\right)}^{\alpha -1}}\right]}^{-1}$$

The built constitutive model was conducted on ZFRP laminates. A quarter finite element model built in [29], which is simply and reveals minimal mesh dependency, was employed. The comparison results were shown in Figure 2. The deviation of the predicted UTS was less than 3%. The simulated failure modes, either matrix tensile failure or matrix shear failure, were also in good agreement with the experimental ones. These verified the applicability of the constructed constitutive model to ZFRPs.

## 4. Simulation Technique for Magnets

### 4.1. Simulation Strategy

The simulation strategy for pulsed magnets was illustrated in Figure 3. Initially, the coupling field of the electrical circuit, magnetic field, and thermal field were solved using a sequential coupling method with a two-dimensional (2D) axisymmetric formulation. The circuit current was firstly solved; then, the calculated current was used to excite the magnet. The induced voltage of the magnet was refreshed by the magnetic penetration method [36]. Next, the Joule heat from the magnetic analysis was loaded on the thermal analysis module and refreshed the coil resistance accordingly [37].

Subsequently, the obtained excited current was applied to perform a 3D magnetic field analysis, utilizing the scalar magnetic potential method. Finally, the 3D Lorentz forces obtained from the 3D magnetic analysis were applied to the structural model, which are represented as
(16)$$\left\{ \begin{array}{l} F_{r}=\int_{V_{node}}\left(\vec{J}\times \vec{B}\right)\cdot e_{r}\cdot dV_{node}\\ F_{z}=\int_{V_{node}}\left(\vec{J}\times \vec{B}\right)\cdot e_{z}\cdot dV_{node} \end{array}\right.$$
where *V_node_* is the effective volume of one single node, while **e_r_** and **e_z_** are the unit direction vector of radial direction and axial direction, respectively. It should be emphasized that the employment of a 2D form for electromagnetic and thermal analysis is used to reduce the computation time, considering the symmetry structure of magnets. The structural analysis was conducted in a 3D form to efficiently carry out a dynamic mechanical examination.

### 4.2. Modeling Method

For the finite-element modeling, the wires and reinforcements were built separately and share an overlapping line on the boundary. Both of them were meshed separately, while contact elements were inserted on the interfaces. Figure 4 shows the mesh of one single conductor layer and corresponding reinforcement. As for the magnetic and thermal analysis, the contact pairs were set as perfect heat dissipation and perfect magnetic contact in the thermal and magnetic analysis. As for the structural analysis, the Coulomb friction model was employed to characterize the interface behavior between the conductor layer and reinforcement, with a friction coefficient of 0.2 [38].

The 2D model present in [36] was employed for the electromagnetic and thermal analysis. A 3D semi-model was used for the structural analysis of the inner magnet to reduce the computing time and enhance convergence, while a 1/2 3D semi-model was used for the outer magnet. The difference is due to the discrepancy ratio of the inside diameter to the height of magnets. The boundary limits in the finite-element modeling included the axial displacement being fixed on the axial cross-section (mid-plane) and the symmetric boundary (*U*_hoop_ = 0) being applied on the hoop cross-section (for the outer magnet). Additionally, since the edges of the magnet are fixed by flanges, the radial and hoop displacement of the end wires were constrained, while the wires could deform freely axially.

The constitutive model constructed in Section 3 was employed to represent ZFRP reinforcements. In regard to wires, the Hill anisotropic plasticity model with kinematic hardening was utilized for Copper–Niobium. The longitudinal and transverse elastic moduli were determined as 78.7 GPa and 11.97 GPa, respectively, while the corresponding yield stresses were 370 MPa and 310 MPa. As for the hard copper wires, an isotropic plasticity model was adopted. The elastic moduli were determined as 42.9 GPa, while the corresponding yield stress was 277 MPa.

### 4.3. Electromagnetic Model Verification

The Lorentz forces are contingent on the magnet’s current and hold a pivotal role in determining the effectiveness of the structural analysis. As a result, a two-layer coil was wound by CuNb and Glass fiber/epoxy to verify the built model, as shown in Figure 5a. This coil consists of 36 turns of wire winding, and the inner bore measures 44 mm. The thickness of the reinforcements are 6 mm and 6.3 mm, respectively. To safeguard against stray magnetic field interference, 30 mm G-10 flanges are employed to secure the inlet and outlet copper bars. The resistance and inductance of the coil stand at 32 mΩ and 80.26 μH, respectively, as measured by a RLC-bridge.

A 1.2 MJ capacitor bank with a capacitance of 3.84 mF was used. The line resistance and inductance were measured at 20 mΩ and 0.6 mH, respectively. During the discharge process, the voltage gradually increased from 6 kV to 9 kV to 15 kV, while the current was monitored using a Rogowski coil. The simulated discharging current closely matched the experimental data, as illustrated in Figure 5b. The primary deviations were observed in the declining portion of the current waveform due to the omission of the line resistance rise. This would not influence the peak current (or the calculated Lorentz forces) and is acceptable.

## 5. Structural Analysis of ZFRPs within Magnets

In this section, analyses were performed on a double-coil magnet system of WHMFC at first. It was originally designed with the von Mises stress failure criterion, and the designed peak magnetic field was 95 T [39]. Unfortunately, this prototype was broken at 83 T, where the stress level is much less than the design limit. Both the inner and outer magnet were destructed, and serious scorch marks were found at the end flanges. Therefore, this prototype was analyzed in detail to determine the reason for the breakage. Subsequently, two successful magnet systems, i.e., the 90.6-T and 94.88-T double-coil magnets of WHMFC, were analyzed in addition. The diameter and thickness of reinforcements within those three magnet were summarized in Table 3.

### 5.1. Inner Magnet Analysis

Analyses of the inner magnet were performed first and incorporated the small deformation formula. Figure 6 presents the stress distribution on the mid-plane at the designed peak magnetic field. ZFRPs in particular exhibited a maximum hoop stress of 2.93 GPa. The maximum axial compression stress in the wire reached 440 MPa, while that in ZFRPs reached 150 MPa. The maximum radial compression stress reached 280 MPa on the interfaces between the wires and ZFRPs, gradually decreasing towards zero in the radial direction. Furthermore, the absence of progressive failure consideration resulted in an approximate 18% increase in the axial compression stress on ZFRPs and a decrease in the hoop stress by 540 MPa. These changes are attributed to the significant variations of the anisotropy of ZFRPs, since the hoop stress increases as the anisotropy of the reinforcements [12].

Although no fiber fracturing was observed, macro-damages occurred on the outer surface of the end of the 2nd–8th reinforcements, about 17.5 turns away from the mid-plane (Figure 7a). It firstly occurred on the outer surface of the 7th ZFRPs when the magnetic field reached a value of 67 T. In addition, the most significant radial stiffness attenuation was observed on the internal surface of ZFRPs (Figure 7b). This can be attributed to the fact that the highest radial compression stress was exerted on the internal surface. The average radial stiffness on the mid-plane measured 2.9 GN/m, with a minimum value of 1.17 GN/m, representing a reduction compared to the original transverse stiffness of 5.29 GN/m. Similarly, the most substantial attenuation in the axial stiffness was observed on the mid-plane (Figure 7c). The average axial stiffness on the mid-plane was 2.81 GN/m, with a minimum value of 2.1 GN/m. In regard to the shear stiffness attenuation (Figure 7d), it was primarily located at the macro-damage position.

Figure 8 serves as a valuable reference to gain insight into the macro-damages. It elucidates that the primary factor contributing to macro-damages on the outer surface of the reinforcement was that the axial tensile stress exceeded the transverse tensile strength of ZFRPs. This axial tensile stress is linked to the forces of the end wires, the direction of which is arctan*B_r_*/*B_z_* against $e_{r}$ (suppose the counterclockwise as the positive direction). Furthermore, Figure 8 reveals that both the inner and outer surfaces on the mid-plane of ZFRPs are susceptible to macroscopic failure, as the stress state closely approaches the failure envelope.

Subsequently, a dynamic structural analysis incorporating the large deformations formula was performed. The rise-time of the pulse was 3.54 ms, and each conductor layer and its corresponding reinforcement (abbreviate this entirety as CR) was analyzed separately. The analysis revealed that the first CR did not undergo any structural damage, while local buckling was observed in the 2nd to 7th CR. Once macro-damages occurred, the failure zone evolved much more rapidly than the statics (Figure 9a–c), ultimately leading to the complete destruction of the entire end-structure (Figure 9d). The end reinforcement could no longer constrain the wires.

Table 4 provides the buckling loads for each CR component. The 7th CR is the most vulnerable part. The predicted failure field exhibited good coincident to the actual numbers. From the aforementioned analyses, it becomes evident that the low transverse strength of ZFRPs is the critical factor leading to macro-damages, posing a significant risk to the magnet system.

### 5.2. Outer Magnet Analysis

The stress distribution of the outer magnet on the mid-plane, calculated under the designed peak magnetic field, is depicted in Figure 10. The maximum axial compression stress in the wires reached 569 MPa, while that in ZFRPs reached 249 MPa. Moreover, the maximum radial compression and hoop stress experienced by ZFRPs were 349 MPa and 2.95 GPa, respectively. Notably, the hoop stress decreased by 470 MPa with the absence of progressive failure, while the axial stress remained relatively unchanged. This negligible change in the axial stress can be attributed to the compression of the CR components within the outer magnet, which could effectively suppress the effect of stiffness attenuation. Furthermore, the maximum axial displacement was about 22% lower compared to the maximum displacement of the proposed model (8 mm). This 8 mm displacement could have severe implications in regard to the insulation.

Figure 11a illustrates the macroscopic failure and stiffness attenuation of ZFRPs at the designated peak magnetic field. No fiber fracture was detected. Macro-damages were observed on the outer surface of the 1st–4th ZFRP layers and the inner surface of the 9th and 10th ZFRP layers. This attributed to the imbalance in the transverse stress, which introduced significant shear stress. The attenuation of the stiffness is depicted in Figure 11b–d. On the mid-plane, the average radial, axial, and shear stiffness measured 1.62 GN/m, 1.35 GN/m, and 1.59 GN/m, respectively.

Finally, a dynamic analysis was conducted. Since the CR components of the outer coil were compressed against each other, no local buckling was observed. The stress distribution resembled closely that of the static analysis with a small deformation.

### 5.3. Other Examples

Two additional magnets, 94.88-T and 90.6-T magnet system of WHMFC, were analyzed in addition. The 94.88-T magnet system maintained the same configurations as the previous 95-T prototype, including the number of conductor layers, the wires in each conductor layer, and the inner bore dimensions. The pulse rise-time of the inner magnet is 3.72 ms and the ZFRP thickness is 3, 3, 5, 5.5, 6.5, 6.7, 7.2, 8 mm. Furthermore, the inner magnet was designed to contribute 55 T at the designed peak magnetic field of 96 T. The accumulated radial and axial Lorentz forces of the 7th CR were 7790 kN and 830 kN at designed peak field, which increased by 5.8% and 19% against the 7th CR of the 95-T prototype. The calculated damaged mode was similar but the calculated failure field increased to 92.9 T. This is mainly because that the ZFRP layers of the 94.88-T magnet system were much thicker.

For the 90.6-T magnet system, main parameters can be found in reference [6]. The pulse rise-time of the inner magnet was 4.8 ms, and the inner magnet was designed to contribute 48.4 T under the peak magnetic field of 90.6 T. The stiffness attenuation of the inner magnet at the designed peak magnetic field is shown in Figure 12. Macro-damages occurred on the outer surfaces of the end of the 4nd–8th reinforcements and the outer surfaces of the 5th–8th reinforcements on the mid-plane. The end damage was due to the axial tensile stress, and the damage on the mid-plane was attributed to the axial compression stress. The axial compression stresses on the mid-plane for the 5th–8th reinforcements measured 145 MPa, 154 MPa, 147 MPa and 149 MPa, respectively, while the radial stresses were zero. This imbalance of the transverse stress would lead to the shear failure mode. The calculated failure field is 89.2 T.

## 6. Discussions

### 6.1. Influence of the Transverse Strength of the Reinforcement

The transverse mechanical properties, denoted as $X_{12}^{0}$, $X_{12}^{u}$, $X_{22}^{t,0}$, $X_{22}^{t,u}$ and $X_{22}^{c}$, were artificially altered to discuss its impact on the magnets’ performance. Both sets of the properties were increased to the same proportion, with the properties of T300/1034-C [29] serving as the target (when the increasing factor reached 1). The corresponding failure magnetic field of the 95-T prototype is shown in Figure 13. Since the Lorentz forces increased quadratically as the magnetic field grew, as depicted in Equation (16) (where *J* is linear to *B*), the failure magnetic field exhibited rapid escalation initially, followed by a deceleration in its growth once the increasing factor reached 0.4. Notably, when the increasing factor reached 0.9, the magnet remained intact. In a word, it is evident that the failure magnetic field increased with the enhancement of transverse properties. This observation underscores the importance of reinforcement materials possessing not only high longitudinal strength but also effective impregnation with epoxy to ensure robust transverse strength.

### 6.2. Influence of the Axial Lorentz Forces

Further to the transverse strength of reinforcements, another focus was the axial Lorentz forces within the magnet, which traditionally have received less attention in design considerations. The axial Lorentz forces were systematically scaled from 0.3 to 1.5 while keeping the radial Lorentz forces unchanged. The resulting angle at which the end wire forces were applied was changed from 5.6° to 26°. The corresponding failure magnetic field is depicted in Figure 14.

The general trend in the failure magnetic field variations can be roughly interpreted as follows: Under a moderate force angle of approximately 23°, the wires predominantly bear most of the axial forces and ZFRPs remain intact. As the axial forces increase further (with the axial force scaling factor exceeding 1.5), the wires approach their plastic limit, and ZFRPs gradually assume a greater share of the axial forces, leading to a reduction in the failure magnetic field. Conversely, when the end force angle is approximately 9°, ZFRPs take on the majority of axial forces, resulting in the lowest failure field. If the axial forces are further reduced, the failure field would increase. It should be emphasized that it is asymmetric to change the axial Lorentz forces in a physical sense; therefore, the trend of the curve in Figure 14 is also asymmetric.

### 6.3. Suitable Reinforcing Material

Previous sections have drawn attention to the limitations of ZFRPs regarding their inadequate transverse strength. This raises concerns about the suitability of Zylon fibers for pulsed high-field magnets. Is there any other fiber that could replace Zylon? Kevlar and other arylon fibers are associated with similar challenges related to insufficient transverse strength as Zylon; Glass fibers have a longitudinal tensile elastic modulus of less than 90 GPa, which is not sufficient to effectively constrain the wire deformation; Carbon and boron fibers do not meet the requirement of electrical insulation. As a result, a synergistic combination of various fibers emerges as the most promising approach. The carbon fiber component should serve as the “sandwich layer”, which is responsible for bearing external loads, while the glass fiber component fully wraps the carbon fiber composites to provide electrical insulation.

Additionally, it is recommended to utilize the wet-winding technique (as opposed to the current semi-wet winding method) to mitigate the splashing of carbon fibers during winding under high pre-stress conditions. Wet-winding can also address the limitations of unsatisfactory impregnation between the fiber and epoxy in semi-wet winding. This approach offers greater flexibility in selecting epoxy materials, without the constraint of viscosity. It allows for the choice of epoxy with the best interfacial properties with the fiber. Furthermore, by employing wet winding in conjunction with the “poly layer assembly” technique [40,41], variable winding pitches of reinforcements, akin to the structure of laminated pressure vessels, can be achieved. The axial strength, stiffness and structural stability could be further enhanced with a carefully designed layup sequence.

## 7. Conclusions

The objective of this study was to investigate the mechanical behavior of ZFRPs reinforcements within high-field pulsed magnets. The study begins with mechanical testing of ZFRPs, followed by the development of its constitutive model, which incorporates the plasticity and progressive damage. The failure modes, damage evolution process, and plasticity deformation of ZFRP can be matters for concern. This model is more advanced than the ideal elastic model, which adopts the maximum equivalent stress failure criterion in the traditional analysis and design of pulsed magnets. Based on this, a comprehensive analysis of the mechanical behavior of ZFRP reinforcements was performed on a failed 95-T prototype from the perspective of composite materials for the first time. The findings revealed a significant attenuation of about 45% in both the radial and axial stiffness of ZFRPs, and the primary cause of failure was identified as local buckling occurring at the end of the inner magnet. Additionally, two successful 90-T magnet systems underwent thorough analyses, and the results exhibited good alignment with the experiments. Finally, the influence of the transverse strength of the reinforcement and axial Lorentz forces on the structural performance of magnets were discussed.

In general, this study provides valuable insight into the composite within pulsed magnets, thereby establishing a solid foundation for further advancements. Future work will focus on the optimization and construction of 100-T-magnet reinforcements with a laminated structure. The response method and multi-disciplinary optimization will be carried out to determine the layup of reinforcements under radial-axial bidirectional Lorentz loads. Then, we will construct a new 100-T magnet with laminated reinforcements using the wet winding and poly layer assembly technique.

## Figures and Tables

**Figure 1 polymers-16-00722-f001:**
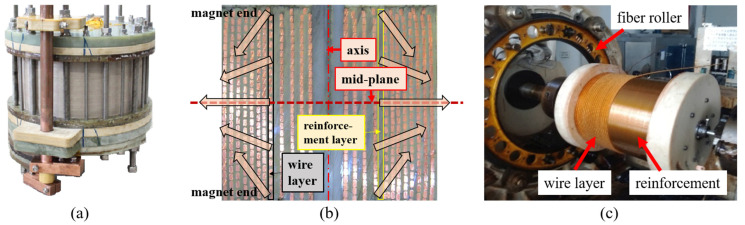
Structure of pulsed magnets and relating winding machine. (**a**) Mono-coil pulsed magnet of WHMFC; (**b**) magnet cross-section (arrow inside indicates the Lorentz forces subjected by conductors); (**c**) customized winding machine of WHMFC.

**Figure 2 polymers-16-00722-f002:**
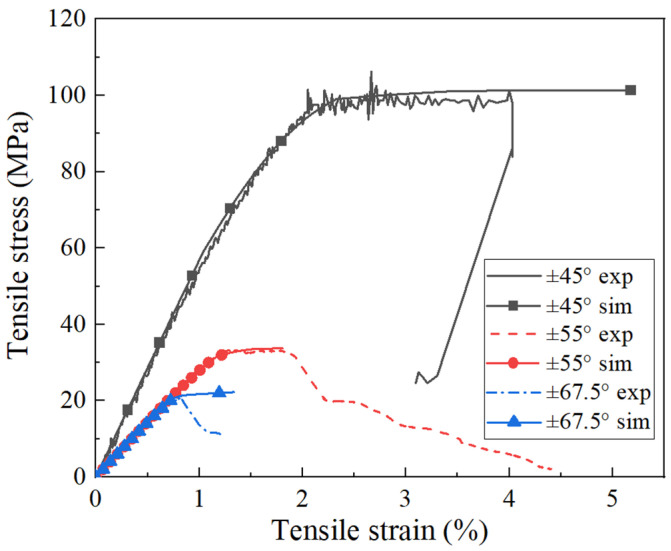
Comparisons between the tensile stress–strain curves of ZFRP symmetric laminates obtained experimentally and by simulation.

**Figure 3 polymers-16-00722-f003:**
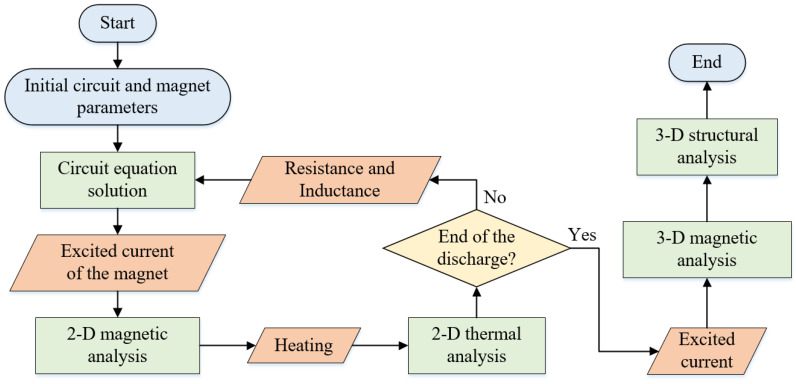
Flowchart of the simulation strategy for pulsed magnets.

**Figure 4 polymers-16-00722-f004:**
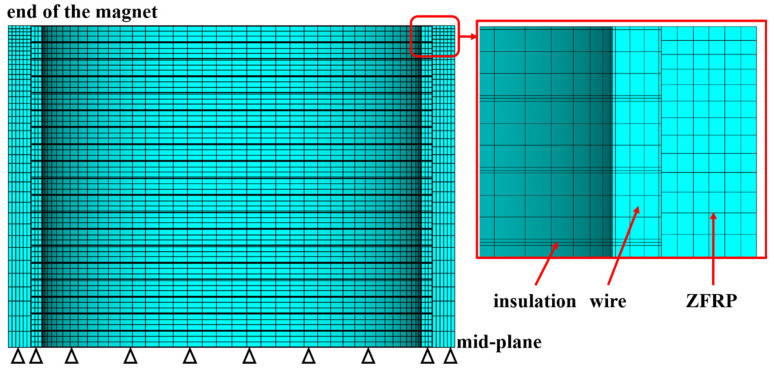
**The** 3D finite-element meshes of pulsed magnets (only shows one conductor layer and corresponding reinforcement).

**Figure 5 polymers-16-00722-f005:**
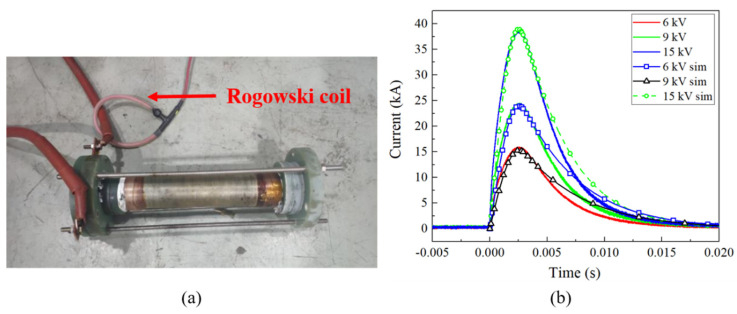
Electromagnetic model verification: (**a**) Experimental configuration; (**b**) Discharging current.

**Figure 6 polymers-16-00722-f006:**
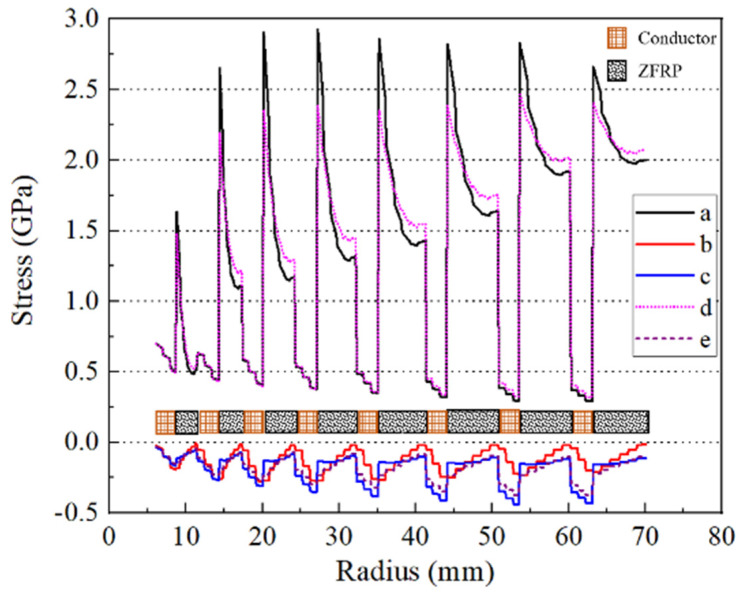
Stress distribution on the mid−plane of the inner magnet. Curves a–c correspond to the hoop, radial, and axial stresses of the proposed model, while curves d and e correspond to the hoop and axial stresses obtained using the ideal elastic model of ZFRPs.

**Figure 7 polymers-16-00722-f007:**
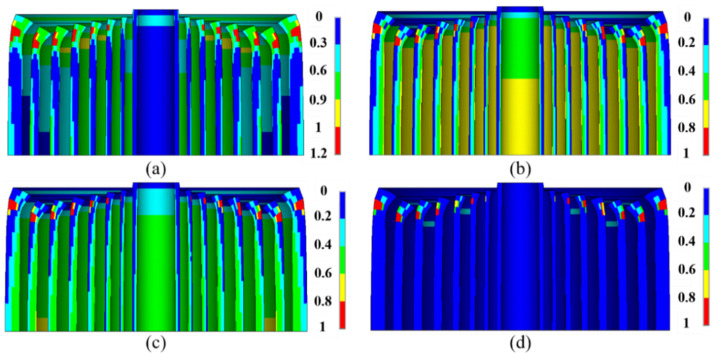
Stiffness attenuation of ZFRP reinforcements of the inner magnet at the designed peak magnetic field. The displacements scale factor is set as 10: (**a**) stress exposure (stress exposure greater than 1 indicated an emergence of macro-damages); (**b**) radial stiffness; (**c**) axial stiffness; (**d**) shear stiffness.

**Figure 8 polymers-16-00722-f008:**
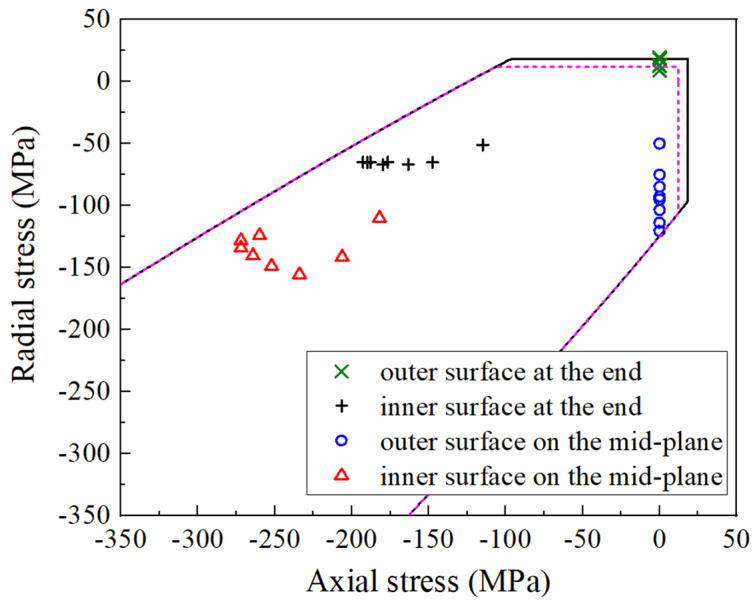
Axial vs. radial stress on the inner and outer surfaces of inner magnet reinforcements. The solid line indicates the macroscopic failure envelope, while the dashed line indicates the micro-failure envelop.

**Figure 9 polymers-16-00722-f009:**
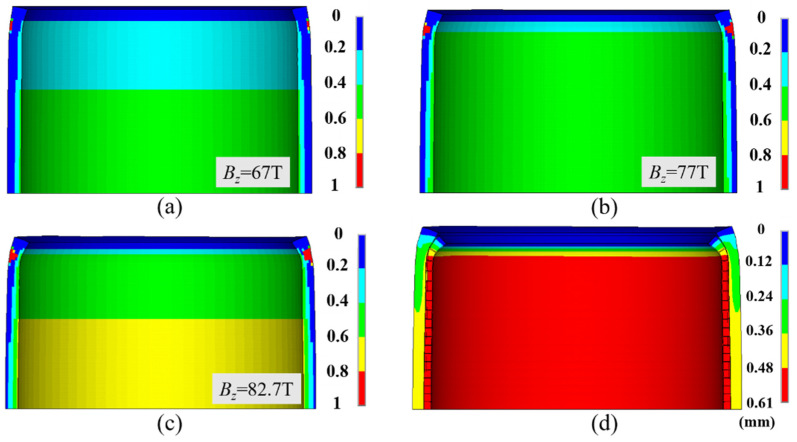
Radial stiffness attenuation of the ZFRP reinforcement of the 7th CR. The displacements scale factor is set as 10: (**a**) radial stiffness attenuation at 67 T; (**b**) radial stiffness attenuation at 77 T; (**c**) radial stiffness attenuation at 82.7 T; (**d**) radial displacement at 82.7 T.

**Figure 10 polymers-16-00722-f010:**
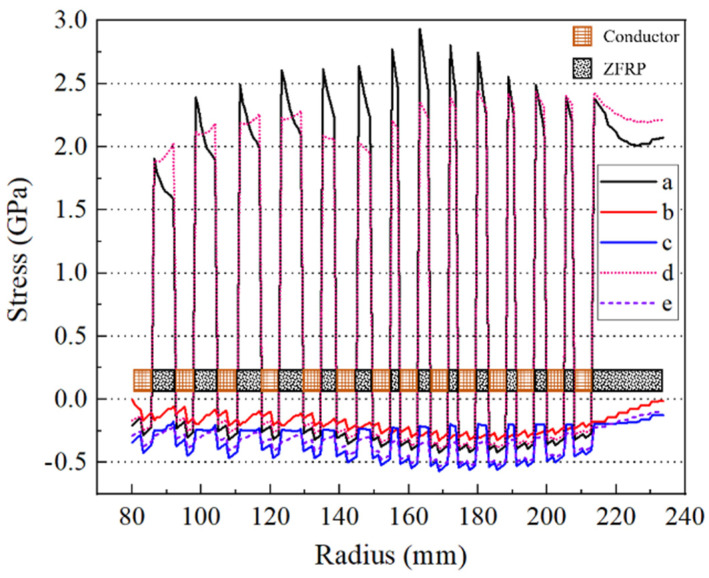
Stress distribution on the mid−plane of the outer magnet. Curves a–c correspond to the hoop, radial, and axial stresses of the proposed model, while curves d and e correspond to the hoop and axial stresses obtained using the ideal elastic model of ZFRPs.

**Figure 11 polymers-16-00722-f011:**
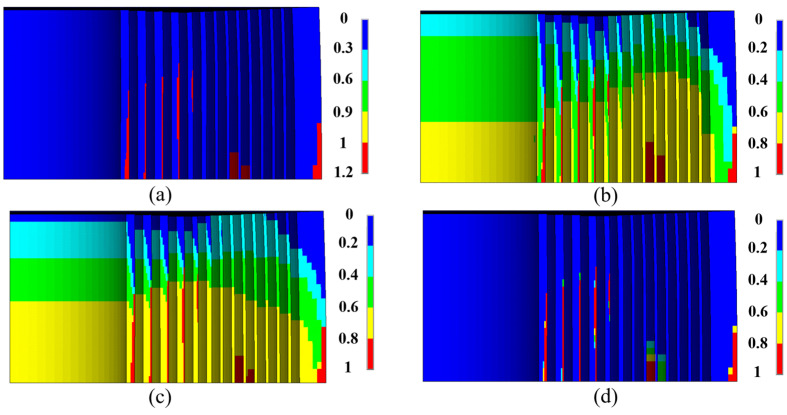
Stiffness attenuation of ZFRP reinforcements of the outer magnet: (**a**) stress exposure; (**b**) radial stiffness; (**c**) axial stiffness; (**d**) shear stiffness.

**Figure 12 polymers-16-00722-f012:**
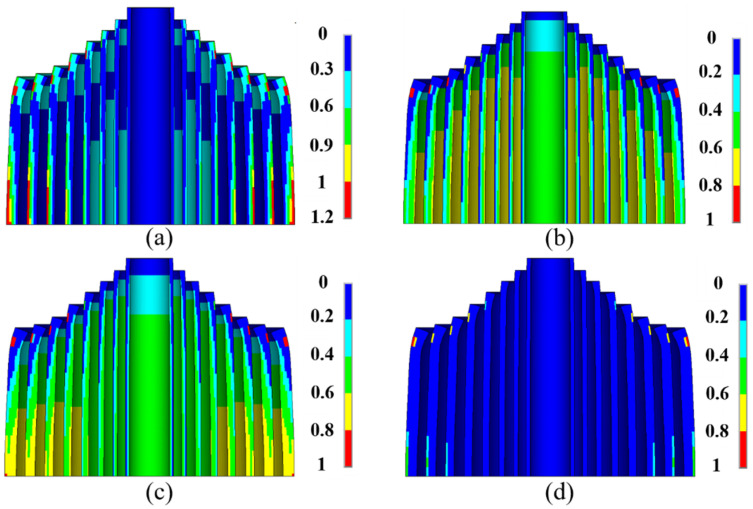
Stiffness attenuation of ZFRP reinforcements of the 90.6-T inner magnet. The displacements scale factor is set as 10: (**a**) stress exposure; (**b**) radial stiffness; (**c**) axial stiffness; (**d**) shear stiffness.

**Figure 13 polymers-16-00722-f013:**
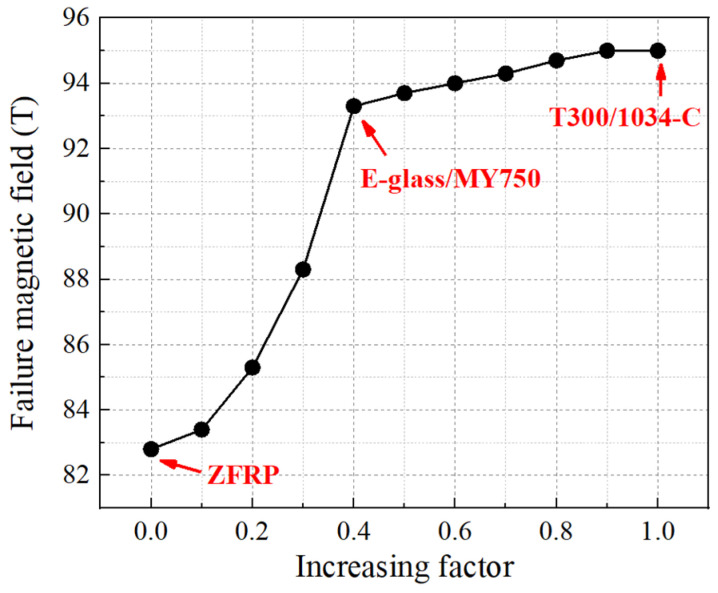
Influence of the transverse properties of reinforcements on the performance of magnets.

**Figure 14 polymers-16-00722-f014:**
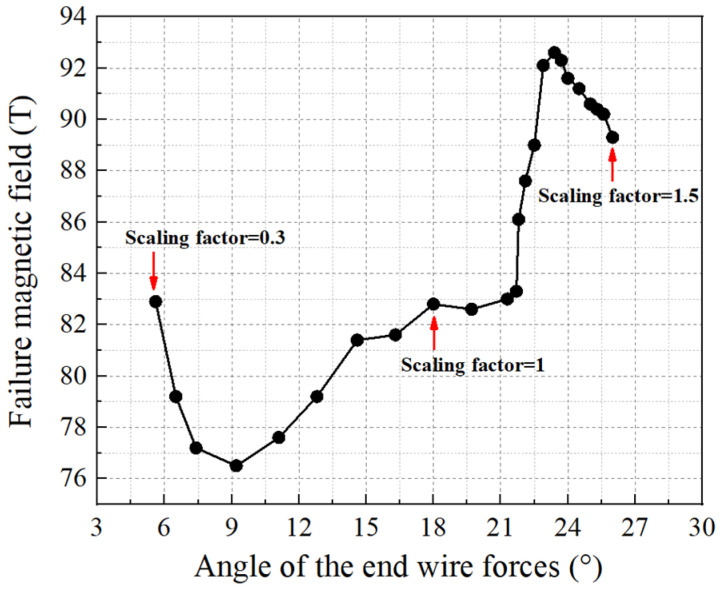
Influence of the axial Lorentz forces on the performance of magnets.

**Table 1 polymers-16-00722-t001:** Mechanical properties of ZFRPs with *V_f_* of 0.53.

Temperature	$E_{11}^{\;t}$ (GPa)	*υ* _12_	$X_{11}^{\;c}$ (MPa)	$E_{11}^{\;c}$ (GPa)	$X_{22}^{\;c}$ (MPa)	$E_{22}^{\;c}$ (GPa)	$X_{22}^{\;t,0}$ (MPa)	$X_{22}^{\;t}$ (MPa)	$E_{22}^{\;t}$ (GPa)	$X_{12}^{\;0}$ (MPa)	$X_{12}^{\;u}$ (MPa)	$G_{12}$ (GPa)
296 K	142	0.36	170	106	81.8	3.66	11.8	18.0	3.12	30.0	49.6	1.49
243 K	/	/	173	110	85.9	3.65	13.8	19.2	3.51	31.2	46.5	1.64
213 K	/	/	/	/	100	4.27	18.5	18.5	3.60	30.6	46.7	1.75
77 K	173.3 [20]	0.36 [20]	170 **	110 **	124.5 *	4.99 *	18.5 **	18.5 **	4.45 *	30.6 **	46.7 **	2.16 *

* Deduced by linear fitting. ** Deduced by extrapolation.

**Table 2 polymers-16-00722-t002:** Mechanical properties of ZFRPs with *V_f_* of 0.8 at 77 K.

*E*_11_ (GPa)	*E*_22_ (GPa)	*G* (GPa)	*ν* _12_	*ν* _23_	$X_{11}^{t}$ (GPa)	$X_{11}^{c}$ (MPa)	$X_{22}^{t,0}$ (MPa)	$X_{22}^{t,u}$ (MPa)	$X_{22}^{c}$ (MPa)	$X_{12}^{0}$ (MPa)	$X_{12}^{u}$ (MPa)
263	3.36	2.06	0.0047	0.6	4.64	170	12	18.3	124	27.3	45.2

**Table 3 polymers-16-00722-t003:** Geometry of analyzed reinforcements.

Parameters	95-T Prototype	90.6-T Magnet	94.88-T Magnet
Number of inner-coil reinforcement layers	8	8	8
Diameter of inner-coil reinforcements (mm)	18.4; 30.2; 42.6; 57.4; 74.2; 93.0; 112.8; 132.6	18.4; 28.8; 40.6; 54.0; 68.8; 84.6; 101.0; 118.4	
Thickness of inner-coil reinforcements (mm)	2.7; 3.0; 4.2; 5.2; 6.2; 6.7; 6.7; 9.0	2.0; 2.7; 3.5; 4.2; 4.7; 5.0; 5.5; 6.0	3.0; 3.0; 5.0; 5.5; 6.5; 6.7; 6.7; 8.0
Number of outer-coil reinforcement layers	14	12	14
Diameter of outer-coil reinforcements (mm)	171.8; 196.6; 221.4; 246.2; 270.0; 289.8; 309.6; 326.4; 343.2; 360.0; 376.8; 393.6; 410.4; 427.2	140.3; 159.1; 177.9; 195.7; 212.5; 227.3; 239.1; 250.9; 262.7; 274.5; 286.3; 298.1	182.8; 206.6; 230.4; 254.2; 277.0; 295.8; 314.6; 330.4; 346.2; 362.0; 377.8; 393.6; 409.4; 425.2
Thickness of outer-coil reinforcements (mm)	6.5; 6.5; 6.5; 6.0; 4.0; 4.0; 2.5; 2.5; 2.5; 2.5; 2.5; 2.5; 2.5; 20.0	5.0; 5.0; 4.5; 4.0; 3.0; 1.5; 1.5; 1.5; 1.5; 1.5; 1.5; 20.0	6.5; 6.5; 6.5; 6.0; 4.0; 4.0; 2.5; 2.5; 2.5; 2.5; 2.5; 2.5; 2.5; 20.0

**Table 4 polymers-16-00722-t004:** Buckling field of each CR component of the inner magnet.

	#1	#2	#3	#4	#5	#6	#7	#8
Buckling magnetic field (T)	-	93.05	93.87	94.86	94.83	92.51	82.76	-

## Data Availability

Data are contained within the article.

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
