# Peer review of "Mechanical Behaviors of Polymer-Based Composite Reinforcements within High-Field Pulsed Magnets"

_polymers, 2024, doi:10.3390/polym16050722_

Round 1
Reviewer 1 Report
Comments and Suggestions for Authors
A manuscript entitled “Mechanical Behaviors of Zylon Fiber‐Reinforced Polymer Reinforcements within High‐Field Pulsed Magnets” has been well structured and written by authors. The manuscript may be submitted after incorporating following points.
· English of low standard authors must uplift standard of writing and remove typo errors.
Authors must enrich introduction by incorporating more recent literature in introduction.
· Authors to advise to clarify the high fiber filling factor (Vf). Does it 0.8 or 0.53.
· Section 4.2: where from authors took the value of COF (0.2)
· What were the boundary limits in FEM?
· Figure 13: the graph was suddenly increased and subsequently it was gradually increased. why?
· Figure 14: the trend of the graph was unsymmetric. why?
· Why did authors not find strain distribution and displacement curve in FEM?
· Authors to advise to mention clearly the diameter of fiber and how much the thickness of filler on magnet spindle.
· Conclusion needs to be revised. Authors must include future scope in this field in last few lines of conclusion.
· Authors must include recent references as only 2-3 references exist from last 2 years.
Comments on the Quality of English LanguageWriting style of English must be improved in revised manuscript.
Reviewer 2 Report
Comments and Suggestions for Authors
The manuscript “Mechanical Behaviors of Zylon Fiber‐Reinforced Polymer Reinforcements within High‐Field Pulsed Magnets” addresses an actual problem related to modeling the performance of polymer composites in strong electromagnetic fields. The study is motivated by the problem of accidental failures of magnets in ultra‐high magnetic fields. The authors focus on the mechanical behaviors of Zylon fiber-reinforced polymers (ZFRPs) within pulsed magnets. Initially, mechanical testing of ZFRPs has been conducted. Then, a constitutive model incorporating the plasticity and progressive damage has been developed. It is shown that a substantial reduction of ~45 % in both the radial and axial stiffness of ZFRPs take place, while the occurred at the end ZFRPs of the inner magnet. In addition, two other industrial magnet systems (90.6 T and 94.88 T) have been simulated. The role of transverse mechanical strength of the reinforcement and axial Lorentz forces on the structural performance of magnets is discussed.
The manuscript falls within the scope of the journal of Polymers.
The state of the art is well characterized. The total number of cited references is equal to 31. However, the self-citation level is ~42% that is unacceptable. It should not exceed 25% (it is by the editors' decision).
The experimental procedure is just modestly explained, while the model and simulation details are deeply described.
The results are duly reported and discussed. There is a Discussion section in the manuscript.
The conclusion lacks numerical outcomes of the calculations; however, they are of importance in the technical study.
The level of English language is OK.
The manuscript requires minor revision. The following aspects are to be addressed by the authors.
The Zylon (IUPAC name: poly(p-phenylene-2,6-benzobisoxazole)) is a trademarked name for a range of thermoset liquid-crystalline polyoxazole. (see wikipedia). In fact, the role of the fibers was not deeply studied. In this regard, the term Zylon might be excluded from the title.
Page 1, line 2. “Fiber‐Reinforced Polymer Reinforcements”. There are too many reinforcements in the title.
Page 1, line 29. “Non‐destructive pulsed magnets”. What does it mean?
Page 1, line 32. “the electronic energy states”. Is this a correct term?
Page 2, line 50. “dislocation of the wires”. Do you mean displacements?
Page 2, line 68. “plastic‐damage coupling behavior”. Is this a correct term?
Page 2, line 81. “plasticity‐damage coupling model”. The same question.
Page 6, figure 2. It is recommended to use the same color-coding for experiments and simulation.
Page 7, line 251. The selection of the friction coefficient of 0.2 is not substantiated.
Page 15. Conclusion. It is recommended to bring more numerical outcomes in order to deeper characterize the obtained results.
Comments on the Quality of English LanguageMinor editing of English language is required.
